# Excimer Laser-Deposited Na_2/3_Ni_1/4_Mn_3/4_O_2_ Film Cathode for Stable Sodium-Ion Battery

**DOI:** 10.3390/nano12173018

**Published:** 2022-08-31

**Authors:** Bailin Lin, Wei Dai, Junhui Tao, Jie Li, Chuanhui Wang, Yang Zhao, Yiping Li, Xinqi Chen

**Affiliations:** 1School of Physics and Mechanical Electrical Engineering, Hubei University of Education, Wuhan 430205, China; 2School of Mathematics and Physics, Jingchu University of Technology, Jinmen 448000, China; 3Hubei Engineering Technology Research Center of Environmental Purification Materials, Institute of Materials Research and Engineering, Hubei University of Education, Wuhan 430205, China

**Keywords:** sodium-ion battery, Na_2/3_Ni_1/4_Mn_3/4_O_2_, discharge capacity, cycling performance

## Abstract

Continued development of lithium-ion batteries is limited by the shortage of Li element. In this situation, the exploration of high-performance sodium-ion batteries is attracting much attention. In this experimental work, Na_2/3_Ni_1/4_Mn_34_O_2_ film cathode materials were fabricated by excimer laser deposition at different oxygen partial pressures. X-ray diffraction studies and field emission scanning electron microscopy revealed high c-axis orientation and uniform grain distribution, respectively, in the deposited films. Furthermore, after 30 cycles under a current density of 13 mA g^−1^, the film samples deposited at an oxygen partial pressure of 65 Pa exhibited a high capacity-retention of 91%. The film structure also had a large-current discharge performance, which makes practical applications possible.

## 1. Introduction

Lithium-ion secondary batteries have the advantages of high energy density, high working voltage, long cycle life, low self-discharge rate, no memory effect, etc., and have been widely used in portable electronic equipment, communication equipment and other devices [1,2,3,4,5]. However, because lithium resources are limited (its abundance on the Earth is only 0.006%), the sustainable application of lithium-ion batteries is restricted. Sodium, which belongs to the first main group with lithium, is abundant and can be easily refined, and the two have similar physical and chemical properties [6,7]. Compared to the lithium-ion battery, the sodium-ion battery has many advantages. First, the standard potential of sodium is 0.3–0.4 V higher than that of lithium; hence, sodium can decompose solvents and electrolyte salts with low potential, and a wider range of electrolyte systems are available for the sodium-ion battery. Second, sodium is more abundant in the Earth’s crust and more evenly distributed. Third, the electrochemical properties of sodium-ion batteries are more stable and safer [8]. Hence, high-performance sodium-ion batteries have important application prospects in low-cost energy storage systems, especially micro batteries, in the future.

The open circuit voltage, discharge-specific capacity and long cycle life of sodium-ion batteries are important indicators of their performance, and cathode materials play an important role in determining these indicators. The cathode material is generally a sodium-containing material that is stable in air and has a high potential. Examples of such materials are the transition metal oxide structure NaMO_2_ (M=Co, Mn, Fe, Ni and other transition metal elements) [9,10,11]. The advantage of the NaMO_2_ structure lies in not only the rich resources and environmental-friendliness of the Mn element but also the multiple crystal forms of the NaMO_2_ structure [12,13]. Among these crystal forms, P2 and O3 are the most typical two-phase structures, and the difference between them is mainly that the sodium ions in the P2 phase are located in a triangular void, whereas those in the O3 phase are located in the oxygen octahedral position. The different positions of Na^+^ results in different reaction mechanisms in the electrochemistry of the two structures [14].

Lee and others found that Na^+^ migrates rapidly in the P2 structure, exhibiting higher reversible capacity and magnification performance than the O3 phase [15]. By introducing the Ni element, both the valence state of Mn and the voltage platform of the material can be improved [16,17]. In 2013, Wang and others prepared P2-Na_2/3_Ni_1/3_Mn_2/3_O_2_ by combining spray drying and solid opposite method and studied its charge and discharge characteristics in different voltage windows [18]. In 2014, Zhao and coworkers were the first to report that the powder electrode of P2-Na_2/3_Ni_1/3_Mn_2/3_O_2_ had a discharge-specific capacity of 150 mAh g^−1^, and the retention rate was 70% after 30 cycles [19].

At present, the cathode material of the Na_2/3_Ni_x_Mn_1−x_O_2_ (x < 1) system has attracted the attention of researchers, and many issues need to be tackled: for example, the change in Ni and Mn atom ratios during performance regulation (previous studies examined Ni and Mn atomic ratios of 1:2), relative to the powder electrode, performance of the Na_2/3_Ni_x_Mn_1−x_O_2_ thin film electrode, and stability of Mn valence state during charging and discharging. In the experiment in this study, Na_2/3_Ni_1/4_Mn_3/4_O_2_ (NNMO) films with a Ni/Mn atomic ratio of 1:3 were grown by a laser deposition method under different oxygen partial pressures, and the structures and morphological characteristics of the thin films prepared under different deposition conditions were compared. Based on this comparison, the Na_2/3_Ni_1/4_Mn_3/4_O_2_ thin film electrodes were designed and their charge–discharge characteristics were discussed.

## 2. Experimental Process

### 2.1. NNMO Target Preparation

The NNMO target was prepared by the traditional solid-phase reaction method, and the raw materials Sodium hydroxide (99.99%), Nickel oxide (99.9%) and Manganese Dioxide (99.9%) were weighed and mixed according to the stoichiometric ratios, with a 50% excess of sodium hydroxide to compensate for the loss of sodium in the subsequent heat treatment process. The raw materials, agate pellets and deionized water were matched according to the mass ratio of 1:3:1.5, and put into the ball mill tank for ball grinding. The ball milling mixture was put into the oven at 80 °C to dry, and ground with a mortar for 1 h to obtain a uniformly mixed raw material. The resulting raw material mixture powder was pre-calcined in a muffle furnace for 24 h and then ground for 1 h, and then calcined for 12 h at 1000 °C.

### 2.2. NNMO Film Growth

Excimer pulsed laser deposition was used to grow NNMO films. The deposition system mainly includes three parts: KrF excimer laser, vacuum chamber and control system. The deposition process is divided into three stages: the target material absorbing the laser energy; the formation of plasma with high-temperature and high-pressure; and expanding of the plasma onto the surface of the substrates. Here the KrF excimer laser has a central wavelength of 248 nm, a pulse width of 25 ns, and a maximum frequency of 50 Hz, so that the laser energy density on the surface of the target is 1.5–3 J·cm^−2^ by adjusting the position of the focusing mirror. Polished stainless steel (SS) is used as the substrate, and the distance between the target and the substrate is fixed at 50 mm. Before deposition, the background vacuum is pumped to 1 × 10^−3^ Pa, and a certain pressure of oxygen (35 Pa, 50 Pa, 65 Pa) is filled when it is deposited; at the time of deposition, the stainless steel substrate is maintained at 750 °C by a temperature controller. Figure 1a shows the plasma feather glow produced by the laser focusing on the surface of the target.

### 2.3. Sodium-Ion Battery Design and Assembly

The thin film grown by laser deposition can be used directly as a working electrode. Sodium sheets (small round slices cut into sodium sheets) are negative electrodes, glass fibers are separators, and 1 M NaClO_4_ (Ethylene carbonate: Propylene carbonate = 2:1 volume ratio) is the electrolyte, assembled into a CR2032 coin cell battery. The entire battery assembly process is done in a glove box filled with argon, and the water and oxygen content is controlled to less than 1 ppm. The battery encapsulation process is shown in Figure 1b: the positive electrode shell is placed in a thin film positive electrode piece, the electrolyte is dripped, the separator soaked in the electrolyte is spread above the positive electrode sheet, and a metal sodium sheet is put in, and then a piece of foam nickel is added as a filler, and the negative electrode shell is pressed tightly.

### 2.4. Sample Performance Characterization

X-ray diffraction (XRD) is used to test the structure, crystallinity, and growth orientation of the prepared NNMO powder and film; Field Emission Scanning Electron Microscopy (FESEM) is used to observe the surface morphology of the sample and the cross-sectional morphology of the thin film sample. The charge–discharge characteristics of the sample are measured by a battery tester (BTS-5V1mA), including magnification characteristics and cycle characteristics.

## 3. Experimental Results and Analysis

Figure 2 is XRD diffractograms of the crystal structure of NNMO films under different oxygen partial pressures. It can be seen that in addition to the peak of the stainless steel substrate, there are only two peaks in the direction of (00l): (002) and (004), indicating that the height along the c-axis along the film is in merit-based orientation. At lower oxygen partial pressures, the diffraction peak is weaker, indicating low crystallinity. This may be due to the low partial pressure of oxygen, which introduces oxygen vacancies. As the partial pressure of oxygen gradually increases, the diffraction peak becomes sharper and sharper, and the crystallinity increases. When the partial pressure of oxygen gradually increases to 65 Pa, the oxygen vacancy is suppressed, the defect is reduced accordingly, and the crystallinity of the film is significantly improved.

Figure 3 shows the surface and cross-sectional FSEM diagram of the NNMO film under different oxygen partial pressures. When the partial pressure of oxygen is 35 Pa, the surface flatness of the film is poor, and a few crystalline particles appear. This may be because the lack of oxygen atoms hinders the growth of the film under low oxygen. As the partial pressure of oxygen rises to 50 Pa, more and more irregular crystalline particles appear, and the boundary between particles is not clear; when the partial pressure of oxygen continues to rise to 65 Pa, the grain size further increases, about 100 nm, showing a relatively uniform nanocrystalline particle. Figure 3d is an FSEM image of the deposited NNMO film cross-section on a SiO_2_/Si substrate. It can be seen that a layer of NNMO film composed of uniformly densely arranged grains is deposited on the surface of the SiO_2_/Si substrate, and the thickness of the film is about 550 nm.

Figure 4 shows the room temperature electrochemical properties of the NNMO film as a sodium-ion battery cathode material under constant current 13 mAg^−1^, voltage window 1.5–4.3 V test conditions. Figure 4a is a constant current cycle curve of the thin film deposited under different oxygen partial pressures. The first discharge-specific capacities of the 35 Pa, 50 Pa, and 65 Pa thin film electrodes were 163.9 mAh g^−1^, 171.1 mAh g^−1^, and 175.3 mAh g^−1^, respectively. It can be seen that, for the samples grown under 35 Pa and 50 Pa, the discharge-specific capacitance decreases rapidly when the number of cycles increases. After 30 cycles, the capacity retention rate was only 48% (78.2 mAh g^−1^) and 63% (108.0 mAh g^−1^) of the initial values, respectively. It may be that under the lower oxygen pressure, the crystallinity of the film is not perfect, oxygen deficiency can form structural defects, the particles are irregular, and the interface is blurred, resulting in slow sodium ion kinetics. When the oxygen pressure gradually increases to 65 Pa, the oxygen defect is inhibited, the crystallinity is enhanced, the crystal plane spacing is increased, and the thickness of the sodium ion layer is also increased, which can enhance the deblocking kinetics of sodium ions. The specific capacity retention rate of the 65 Pa thin film electrode is 91%, which is much higher than that of the 35 Pa and 50 Pa thin film electrodes, showing excellent cycle stability.

The first charge–discharge curve of the NNMO thin film electrode is shown in Figure 4b. When the partial pressure of oxygen is 35 Pa and 50 Pa, the curve is smooth and sloped, and the voltage platform is almost invisible. It may be that the hypoxia and incomplete crystalline state of the thin film material under low oxygen pressure make it difficult to react redox or phase change during charge–discharge so that no charge–discharge platform appears. With the increase of oxygen partial pressure, it can replenish the oxygen content that is missing due to high temperature sputtering during film growth, and the crystallinity is also improved. When the partial pressure of oxygen is increased to 65 Pa, three obvious voltage platforms appear on the charge–discharge curve, located around 4.0 V, 3.5 V, and 2.0 V, respectively. The voltage platform at 4.0 V here is caused by the P2-O2 phase transition, and the voltage platform at 3.5 V and 2.0 indicates that the two pairs of redox reactions^12^ have occurred, Ni^2+^/Ni^4+^ and Mn^3+^/Mn^4+^, respectively [20,21].

Figure 5 compares the discharge rate characteristics of the NNMO thin film electrode under different oxygen partial pressures, and the charge and discharge voltage range and charging current density are the same as the previous parameter settings. Obviously, in a lower oxygen partial pressure environment (35 Pa and 50 Pa), the thin film material is in a mixed state of crystalline and amorphous state, and this incomplete crystallization during charge–discharge will hinder the diffusion and migration rate of sodium ions and electrons, so the first discharge specific capacity is low, and the decay is faster with the increase of discharge current density. When the partial pressure of oxygen is increased to 65 Pa, the crystallinity of the thin film material is improved, the particle size is uniform, and the discharge-specific capacity and capacity retention rate are improved. Although it also shows rapid decay at higher current densities above 520 mA g^−1^, when the current density is restored to 130 mA g^−1^, its discharge-specific capacity can be maintained at about 70%, indicating that the thin film cathode material can withstand high current charge and discharge.

The above experimental results show that the NNMO thin film cathode material with oxygen partial pressure deposited at 65 Pa shows better battery characteristics. Based on the X-ray diffraction and SEM analysis, this enhancement can be ascribed to the following reasons. First, the thin film with better crystallinity can improve the electron conductivity of the active material as well as the transfer rate of charge. Second, the uniform-distributed grains with less defects are more conducive to the migration of ions and electrons, which can shorten the transmission distance and improve the circulation stability of the electrode.

## 4. Conclusions

In summary, the NNMO target was sintered by the traditional solid-phase reaction method, and NNMO films with high c-axis optimization orientation and uniform grain distribution were obtained on the surface of stainless-steel substrates by excimer laser deposition technology. When the partial pressure of oxygen was 65 Pa, the NNMO thin film samples exhibited greatly improved sodium storage performance and had the higher discharge-specific capacity and better cycle characteristics and magnification performance; after 30 cycles, the discharge-specific capacity was maintained at 91% of the first discharge-specific capacity at a current density of 13 mA g^−1^, reaching 159.5 mAh g^−1^. After applying a high current density (e.g., above 520 mA g^−1^) and then restoring it to a lower value (e.g., 130 mA g^−1^), its discharge-specific capacity was still maintained at about 70%, with a strong ability to quickly de-embed. Through the optimization of doped elements, atomic ratios and film growth parameters, the performance of the NNMO thin film cathode materials can be further improved.

## Figures and Tables

**Figure 1 nanomaterials-12-03018-f001:**
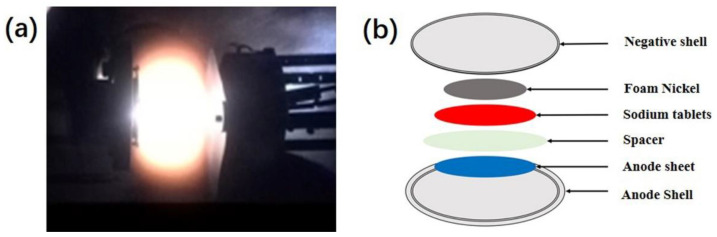
(**a**) Plasma plume generated by laser sputtering target; (**b**) Assembly diagram of cathode.

**Figure 2 nanomaterials-12-03018-f002:**
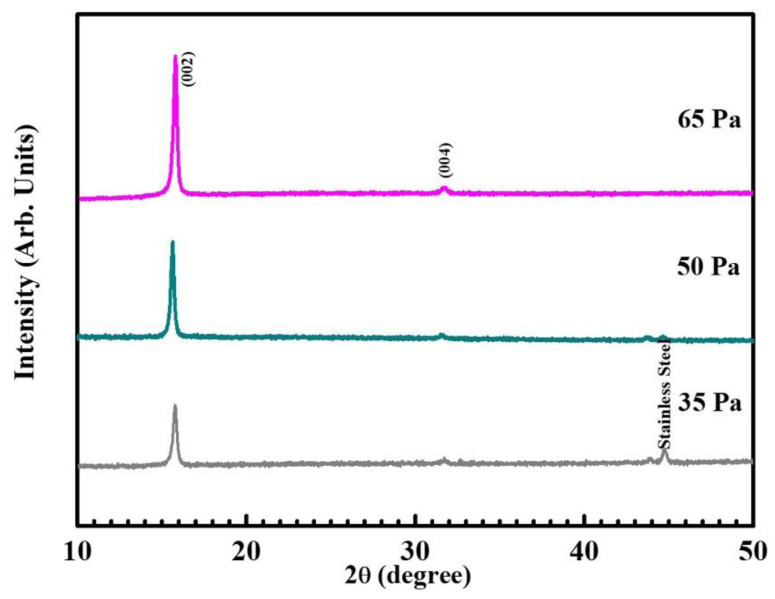
XRD diffraction patterns of NNMO films at different oxygen pressures.

**Figure 3 nanomaterials-12-03018-f003:**
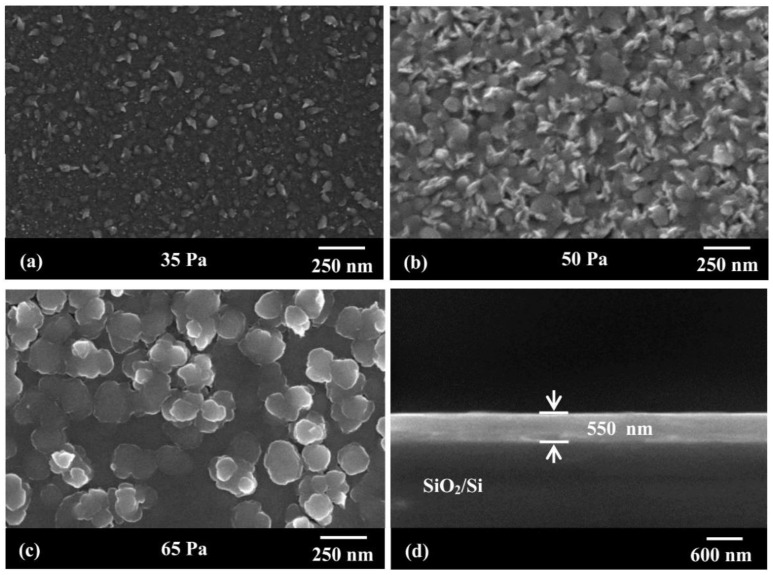
Surface FSEM morphology of NNMO films deposited at different oxygen pressures (**a**) 35 Pa, (**b**) 50 Pa, (**c**) 65 Pa, (**d**) cross-sectional view of the films deposited on SiO_2_/Si substrates.

**Figure 4 nanomaterials-12-03018-f004:**
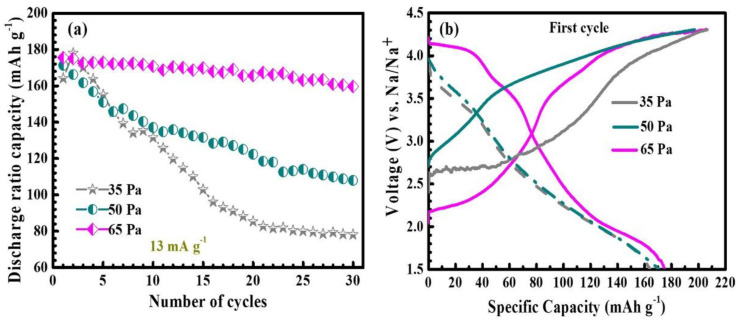
(**a**) cycling curves of NNMO thin film electrode materials at different oxygen pressures; (**b**) first charge/discharge curves (constant current 13 mA g^−1^, voltage window 1.5–4.3 V).

**Figure 5 nanomaterials-12-03018-f005:**
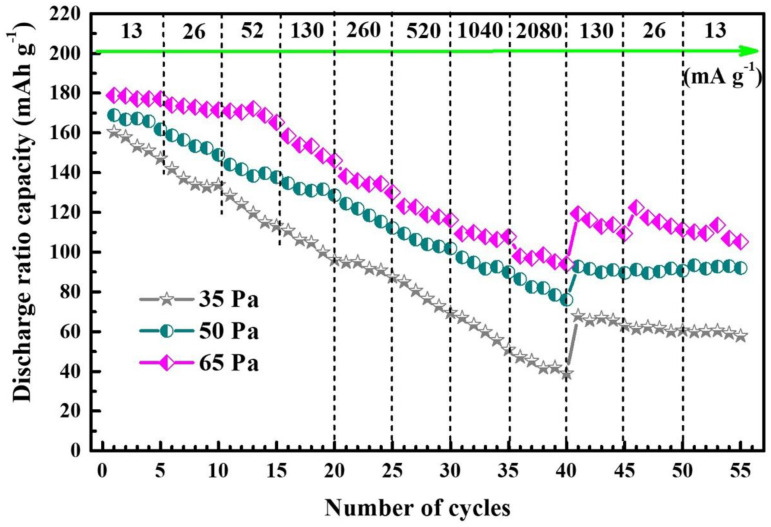
Cycling curves of NNMO thin film electrode materials discharged at different current densities.

## Data Availability

The data presented in this study are available on request from the corresponding author.

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
