# Peer review of "Excimer Laser-Deposited Na2/3Ni1/4Mn3/4O2 Film Cathode for Stable Sodium-Ion Battery"

_nanomaterials, 2022, doi:10.3390/nano12173018_

Round 1

Reviewer 1 Report

This manuscript describes the excimer laser deposition of Na2/3Ni1/4Mn3/4O2 layered cathode for sodium-ion batteries. Overall, the thin film strategy introduced in this manuscript benefits scientific society, making this manuscript publishable to this journal. However, I found that the following comments are necessary to improve the scientific value of this manuscript:

1. This manuscript controls the partial pressure of oxygen gas during the deposition. The highest performances were obtained for the highest partial pressure of 65 Pa; this necessitates further investigation at increased partial pressure to see if further enhancement is possible there.  

2. The manuscript requires English editing service; there is a grammatical error from the first sentence of the abstract. 

3. Conclusion part needs improved structure; for example the first sentence of the present conclusion is not adequate for the opening of the paragraph. 

Author Response

Thank you for the reviewer’s comments. We have made modifications one by one according to your opinions. Please see attached file.

Reviewer 2 Report

Please, find the attached report 

Author Response

(The authors gave the same response as above.)

Round 2

Reviewer 1 Report

The manuscript was corrected according to the comments.